# TUH-NAS: A Triple-Unit NAS Network for Hyperspectral Image Classification

**DOI:** 10.3390/s24237834

**Published:** 2024-12-07

**Authors:** Feng Chen, Baishun Su, Zongpu Jia

**Affiliations:** 1School of Computer Science and Technology, Henan Polytechnic University, Jiaozuo 454003, China; subaishun@hpu.edu.cn; 2School of Software, Henan Polytechnic University, Jiaozuo 454000, China; jiazp@hpu.edu.cn

**Keywords:** hyperspectral image classification, triple unit, neural architecture search, hybrid search space, attention mechanism

## Abstract

Over the last few years, neural architecture search (NAS) technology has achieved good results in hyperspectral image classification. Nevertheless, existing NAS-based classification methods have not specifically focused on the complex connection between spectral and spatial data. Strengthening the integration of spatial and spectral features is crucial to boosting the overall classification efficacy of hyperspectral images. In this paper, a triple-unit hyperspectral NAS network (TUH-NAS) aimed at hyperspectral image classification is introduced, where the fusion unit emphasizes the enhancement of the intrinsic relationship between spatial and spectral information. We designed a new hyperspectral image attention mechanism module to increase the focus on critical regions and enhance sensitivity to priority areas. We also adopted a composite loss function to enhance the model’s focus on hard-to-classify samples. Experimental evaluations on three publicly accessible hyperspectral datasets demonstrated that, despite utilizing a limited number of samples, TUH-NAS outperforms existing NAS classification methods in recognizing object boundaries.

## 1. Introduction

Hyperspectral image (HSI) classification technology is of paramount importance in the realm of remote sensing, as it captures and analyzes spectral information from visible light to the near-infrared spectrum, thereby enabling precise identification of terrestrial objects. With the advent of deep learning algorithms, the performance of HIS classification has seen remarkable improvement, allowing researchers to delve deeply into the spatial and spectral characteristics embedded within spectral data, thus propelling the ongoing advancement of remote sensing technology. 

Early research on the classification of HSIs focused on spectral feature extraction and widely used methods such as principal component analysis, the envelope removal method, and the spectral angle mapper algorithm. The extracted features were subsequently input into various supervised classifiers, such as logistic regression, naive Bayes, and artificial neural networks, to achieve effective classification. However, relying solely on spectral information for hyperspectral classification presents numerous challenges. For instance, the Hughes phenomenon indicates that the demand for training samples increases sharply with the number of bands. This leads to a scenario where classification accuracy first improves and then declines when samples are limited. Additionally, HSIs often contain mixed pixels, in which the spectral characteristics arise from the reflections of multiple objects, making it more likely that classification errors will occur if only spectral information is considered.

Researchers have begun to fuse spectral features with spatial features for HSI classification to overcome the limitations of single-dimensional information. Deep learning methods have gradually replaced traditional HSI classification methods in this context. Convolutional neural networks (CNN) in HSI classification have evolved from using spectral dimensions to spatial dimensions and then to space-spectrum associations. At first, 1D CNN [1] was used to focus on extracting spectral features. However, spectral information alone is insufficient for the HSI classification task to achieve accurate results. As a result, 2D CNN [2] was proposed to extract spatial information. Nevertheless, neither 1D nor 2D CNN takes full advantage of the three-dimensionality of HSI. Consequently, 3D CNN [3] has been applied in HSI classification to achieve a comprehensive fusion of HSI’s spatial spectral features. During this period, Dosovitskiy et al. presented the vision Transformer (ViT) [4] model, which effectively alleviates the receptive field limitation issues faced by traditional convolutional approaches through its multi-head attention mechanism. Since then, many methods combining ViTs and CNNs have appeared. SpectralFormer [5] was the first to apply the Transformer to the HSI classification task. SSFTT [6] integrated a backbone CNN and Transformer, with convolutional layers used to capture low-level features and the Transformer used to generate high-level features. GAHT [7] presented a grouped pixel embedding module that restricts multi-head attention to the local spatial-spectral domain, allowing for HSI classification from a spatial-spectral perspective in a global-local manner. DBSSAN [8] is a dual-branch spectral-spatial attention network, where the spatial branch fuses global and local spatial features through the proposed spatial self-attention module. In contrast, the spectral branch utilizes the Transformer model to extract spectral features. The extracted features are then fused and classified using a multi-layer perceptron. These methods not only inherit the global feature perception capabilities of the ViT model but also retain the strengths of CNNs in local feature fusion.

Accurate feature extraction can enhance the model’s performance and reduce computational complexity. Many studies have focused on improving the accuracy and diversity of spatial-spectral feature extraction. MASSFormer [9] is a memory-augmented spectral-spatial Transformer that introduces a memory tokenization module to convert spectral-spatial features into memory tokens, preserving local context and significant features, thereby enhancing the model’s feature representation capability. The memory-augmented Transformer encoder achieves sufficient information mixing through an expanded multi-head self-attention mechanism, improving the performance of HSI classification. Wang et al. [10] constructed a comprehensive feature extraction framework by combining global feature extraction, local feature extraction, and feature alignment strategies. This framework can fully utilize the spectral and spatial information of HSIs to extract more representative and discriminative features, providing strong support for classification tasks. M3FuNet [11] is an unsupervised multi-feature fusion network that extracts spectral and spatial features through multi-scale supervector matrix correction and multi-scale random convolutional dispersion methods and achieves feature calibration through feature fusion and decision fusion, enhancing the performance of HSI classification. APFL [12] is a semi-supervised adaptive pseudo-label feature learning model that improves the accuracy of HSI classification by utilizing unlabeled sample information through iterative multi-scale super-pixel segmentation and pseudo-label feature generation. Therefore, compared to methods that rely solely on CNNs to obtain spectral-spatial features, these hybrid methods can extract more comprehensive and richer spectral-spatial features.

While achieving good results, the hyperspectral classification methods mentioned above also have several challenges. One major issue is the rigidity of existing neural networks, which cannot adapt to the complexity and diversity of HSI data. When processing HSI data, neural networks must possess adaptable and flexible receptive fields to accommodate the diverse sizes and shapes of surface features [13]. The spectral-spatial asymmetry inherent in HSIs must be effectively managed. Furthermore, given the pronounced differences such as spatial resolution, spectral coverage, and the number of bands among various HSI datasets, researchers must thoroughly consider and address these issues when designing neural networks, as they often present difficulties, making the design process cumbersome and often limited by the designer’s experience. NAS [14] is an automated approach for creating neural network structures that reduces the complexity of the design cycle and its reliance on expertise compared to manually customizing the structure for each dataset. 

Current NAS methods for HSI classification are categorized into global and cell-based search spaces [15,16]. Global research involves constructing a directed acyclic graph to identify an optimal set of operators, but this often results in numerous candidate architectures, consuming considerable computational resources and time. In contrast, cell-based search spaces offer flexibility in managing the number of cells tailored to specific scenarios and come in two forms: one utilizing a single cell type that views hyperspectral data holistically and another using two separate cells for spectral and spatial data. While the former acknowledges the relationship between spectral and spatial dimensions, it underestimates the importance of spectral data in classification. The latter method emphasizes feature extraction but overlooks the interdependency of spectral and spatial information, which is crucial for distinguishing categories with similar spectral characteristics, especially in complex environments. Moreover, prevalent patch-based input strategies in existing deep learning models [17] can lead to a loss of global contextual information, undermining the contribution of spatial data to classification tasks. Thus, there is a pressing need for methods that effectively integrate spectral and spatial dimensions to fully leverage the unique characteristics of hyperspectral data.

In this paper, we propose a new triple-unit hyperspectral NAS network named TUH-NAS to address the main issues present in cell-based search methods. TUH-NAS features an inner search space comprising a spectral processing unit (SPEU), a spatial processing unit (SPAU), and a feature fusion unit (FFU). Each unit has distinct functions, with SPEU focusing on deep spectral information mining, SPAU on spatial feature extraction, and FFU on merging information from the first two units to strengthen the link between spectral and spatial dimensions. Additionally, the model incorporates an HSI attention mechanism module (HSIAM) in the search space to increase sensitivity to crucial features, such as edge regions. To ensure effective training, we utilized a comprehensive loss function that facilitates balanced objectives for accurate classification while emphasizing edge significance. The FFU specifically addresses patch-based input patterns, enabling the model to concentrate on localized details and subtle spectral variations, ultimately improving accuracy in identifying edge land cover targets.

The primary contributions presented in our paper are as follows:This paper proposes a triple-unit hyperspectral NAS, which includes three types of processing units: the spectral processing unit, the spatial processing unit, and the feature fusion unit. The entire search space can be divided into internal and external spaces.We designed an attention mechanism module named HSIAM, specifically tailored for HSI data processing. This module integrates multi-scale feature fusion and enhanced channel technology to improve the spectral, spatial, and channel feature representation capabilities of the input feature maps. Additionally, we developed a triple fusion loss function that comprehensively considers both spectral and spatial information, capturing feature differences and similarities from multiple aspects, thereby enhancing classification accuracy.The outcomes of experiments on the MUUFL, Houston2018, and XiongAn datasets indicate that our methodology improves the overall accuracy by approximately 3%, 5.6%, and 3.5% compared to the baseline methods, demonstrating that the proposed method significantly enhances the precision of HSI classification methods.

The remainder of this paper is structured as follows: Section 2 offers an overview of related research. Section 3 discusses our methodology in detail. Section 4 describes the experiment setup and evaluates the results. Finally, in Section 5, we summarize our findings.

## 2. Related Work

### 2.1. Neural Architecture Search

NAS is a method for automated machine learning to discover optimal neural network architecture for specific tasks, thereby reducing the need for manual design and tuning. Early NAS methods in HSI classification primarily relied on evolutionary algorithms [18] or reinforcement learning [19] to explore various network architectures. However, these methods often require substantial computational resources due to the necessity of training and evaluating numerous candidate networks. Over the past few years, some new HSI classification methods using NAS technology have emerged. To illustrate, Zhang et al. [20] suggested a method grounded in particle swarm optimization for architecture search focused on a unit-based CNN to classify HSIs. Wang et al. [21] proposed a lightweight deep-learning network named LMAFN for the classification of HSIs. AAtt-CNN [22] combined NAS technology with a channel attention mechanism, enabling the automatic design and optimization of 1D and 3D CNN architectures suitable for HSI classification. Xiao et al. proposed HCFSL-NAS [23], which automatically utilizes differentiable NAS to search for the optimal embedding feature extractor and employs a multi-source learning framework to aggregate rich heterogeneous and homogeneous source data. This allowed the network to attain satisfactory classification results with just a limited number of labeled samples.

Recently, gradient-based NAS, particularly differentiable architecture search (DARTS) [24], has gained increasing attention. This method seeks optimal network architectures by constructing a directed acyclic graph (cell) comprising N-ordered nodes. At the heart of DARTS lies the transformation of the NAS dilemma into a differentiable problem, thereby enabling the utilization of gradient descent for the optimization of architectural parameters. Not long after DARTS was proposed, researchers commenced their exploration into the application of automated CNNs, utilizing grounded-in gradient descent methodologies, for HSI classification. Chen et al. [25] were pioneers in employing automated CNNs for HSI classification, designing both 1D and 3D Auto-CNN frameworks tailored for spectral and spectral-spatial classification, respectively. Subsequently, a plethora of search strategies emerged. Liu et al. [26] proposed CPSO-Net, which converts the NAS architecture into particles, utilizing particle swarm optimization algorithms to identify optimal structures. Hu et al. [27] developed I-NAS, which inputs entire HSIs with label masking, employing gradient search techniques to ascertain the optimal unit. Song et al. [28] designed MS3ANAS, enhancing the structural vector via an expanded multi-scale attention mechanism coupled with a slow-fast learning strategy for optimization. Wang et al. [29] introduced the CK-βNAS-CLR framework, which incorporates β decay and confidence learning rate strategies to mitigate discretization discrepancies during the search process. Cao et al. [30] proposed SCIFNAS, which integrates sparse coding and feedback mechanisms to form a search space dedicated to HSI classification. The focus of these methods is primarily on researching NAS architecture search strategies. At the same time, hyperspectral data are generally processed in their entirety within the framework without distinguishing between spectral and spatial dimension information.

As research deepens, researchers recognize that processing spectral and spatial information separately helps improve hyperspectral classification performance [31,32]. Therefore, researchers have begun experimenting with deploying different cell unit types within search frameworks to handle spectral and spatial dimension information individually. Zhang et al. [33] proposed a classification model called 3D-ANAS. This model features a three-dimensional asymmetric search space that utilizes different decomposed convolutions to handle spectral and spatial features. Xue et al. [34] proposed a hybrid CNN-Transformer framework called Hyt-NAS. They designed a new hybrid search space that includes spatial-dominant and spectral-dominant units and attempted to integrate Transformer modules into automatically designed convolutional neural networks to enhance classification accuracy. Zhan et al. [35] proposed U^2^ConvFormer for HSI classification, which combines a nested U-Net with a scale-aware Transformer. They designed an asymmetric spectral-spatial convolution, where the asymmetric spectral-spatial feature pooling can separately pool spectral and spatial information. 

### 2.2. Attention Mechanism in HIS Classification

In the realm of HIS classification, attention mechanisms have emerged as an indispensable component. A sophisticated attention mechanism can significantly enhance feature extraction capabilities, enabling the model to concentrate more acutely on pivotal features. In HSI classification tasks, the commonly used attention types are spectral and spatial attention. RS-AMCNN [36] proposed a branch attention mechanism, where each branch processes different spatial window inputs and predicts the category of that spatial window. This approach emphasizes more discriminative branches while suppressing less useful ones. Alkhatib et al. [37] presented a dual-branch model that leverages sophisticated feature fusion networks combined with an attention mechanism to enhance classification accuracy. They underscored the pivotal role that the attention mechanism plays in the analysis of HSI. Ayuba et al. [38] addressed the inefficient utilization of hyperspectral data in deep learning models by introducing a self-supervised contrastive learning network called HyperKon, demonstrating the potential of attention mechanisms in improving classification results. Kang et al. [39] presented a dual-branch spectral-spatial network that incorporated an attention mechanism for HSI classification, aiming to achieve high accuracy while minimizing the number of model parameters. Wang et al. [40] designed a mechanism known as CNN-enhanced Cross Attention (TNCCA), which leverages multi-scale hyperspectral input data and amalgamates multi-scale 3D and 2D hybrid CNNs to extract superficial spatial-spectral features. This approach achieved good classification results.

In the abovementioned methods, spectral and spatial attention are used to extract features. Still, they are separated and processed through different network branches, which are either sequential or in parallel. This approach lacks joint attention information, resulting in ineffective feature interaction, failure to leverage spatial information to enhance spectral features, and a lack of utilization of spectral information to improve spatial features. Such outcomes are unfavorable for edge recognition.

## 3. Proposed Method

The main objective of the proposed TUH-NAS is to enhance the integration of spectral and spatial data, aiming to achieve greater accuracy in HSI classification tasks. In this section, we will commence by presenting the comprehensive framework of TUH-NAS, encompassing both its internal and external search architectures. Then, we introduce the network’s attention mechanism module and loss function, respectively.

### 3.1. The Overall Architecture of TUH-NAS

TUH-NAS provides a complex NAS model for HSI classification tasks. This model integrates a new hybrid search space to handle HSI data’s spectral and spatial information effectively. The process of NAS can be divided into two components: an internal search and an external search. The internal search strategy determines the internal topology of each working unit, while the external search identifies the type of working unit for each layer. The overall algorithm flowchart of the proposed method is shown in Figure 1.

#### 3.1.1. Internal Search Architecture

The internal search architecture of TUH-NAS is illustrated in Figure 2a. Each computational unit comprises multiple nodes that receive two inputs from the current unit along with outputs from all preceding nodes. These nodes process the information via multiple pathways, each corresponding to a set of candidate operations. Each operation is accompanied by a weight that can be fine-tuned using a gradient descent algorithm. Ultimately, each node retains the top two most effective paths with the highest weights, thus establishing a fixed fundamental cell, as depicted in Figure 2b. It is assumed that each path in the cell contains C candidate operations, and the output of node *N_i_* is
(1)$$n_{i}=\sum_{x=1}^{C}\left(\rho_{i}^{x}\cdot \theta_{i}^{x}\right)$$
where *ρ* denotes different candidate operations and *θ* represents their corresponding weights, which are derived through self-learning. The output of the work unit $h_{l}^{j}$ is obtained as follows:
(2)$$h_{l}^{j}=conv\left(concat\left(n_{i}^{j}\right)\right)$$
where *j* indicates the unit type, *l* signifies the layer number, *i* ∈ {1, 2, …, *N*}, and *N* is the number of nodes within each fundamental unit.

Each work unit contains nine dedicated candidate operations and two common operations, resulting in twenty-nine candidate operations within the entire architecture, higher than other NAS methods. When using NAS for HSI classification tasks, a more diverse set of candidate operations allows the network to achieve more complex feature extraction, benefiting the identification of land cover edges. The candidate operations included in each unit are shown in Table 1.

The meanings of the various candidate operations are as follows:
econ_*i* − 1: LReLU − Conv(*i* × 1 × 1) − BN.
esep_*i* − 1: LReLU − Sep(*i* × 1 × 1) − BN.
acon_1 − *i*: LReLU − Conv(1 × *i* × *i*) − BN.
asep_1 − *i*: LReLU −Sep(1 × *i* × *i*) − BN.
con_*i* − *j*: LReLU −Conv(1 × *j* × *j*) − Conv(*i* × 1 × 1) − BN.
dilated_*i* − 1: Dilated convolution with a kernel size of *i* × 1 × 1 and a dilation factor 2.
dilated_1 − *i*: Dilated convolution with a kernel size of 1 × *i* × *i* and a dilation factor 2.dilated_*i* − *i*: Dilated convolution with a kernel size of *i* × *i* × *i* and a dilation factor 2.
skip_connect: *f*(*x*) = *x*.
none: *f*(*x*) = 0.
where LReLU denotes the Leaky ReLU activation function; BN refers to batch normalization; Conv signifies standard convolution; and Sep represents separable convolution.

#### 3.1.2. External Search Network

In the TUH-NAS, each network layer comprises three working units: a spectral processing unit, a spatial processing unit, and a feature fusion unit. The external search network determines which kind of working unit will be selected for the final trained network. The external architecture of the search network is depicted in Figure 3.

While conducting topological structure research within the working units, we also perform computations on the external outputs of these units. During the external architecture search process, the SPEU and SPAU of each network layer simultaneously extract features from the spectral and spatial dimensions of the input image, respectively. The outputs from SPEU and SPAU are then used as input for the FFU to integrate spectral and spatial data. An attention mechanism module, HSIAM, is connected after the FFU to focus on critical spectral-spatial regions. Thus, the output $h_{l}$ for layer l generated by the HSIAM can be represented as:(3)$$h_{l}=HSIAM(FFU(S_{SPEU},\;S_{SPAU})$$

To quantify and regulate the contribution of each working unit in the final decision-making process, we assign a weight coefficient γ to each working unit. This coefficient reflects the importance of each unit’s output in the overall decision and provides a flexible mechanism for adjusting the network architecture based on task requirements. For each network layer, we select the working unit with the highest weight γ as the representative for that layer, which will participate in constructing the final optimized training network. By assigning different initial values to the weight γ, we can effectively intervene in the external search process, allowing for tailored architecture search schemes for various application scenarios. Figure 4 depicts the architecture of the optimized training network.

At the end of optimizing the training network, we connected a standard Transformer module, which works in conjunction with the HSIAM in the network. This combination can further enhance the ability to capture global contextual information, allowing the model to learn long-range dependencies between input features. The attention mechanism helps the network selectively amplify important information at different resolution levels, while the Transformer further enhances the interactions among features through self-attention, enabling the network to learn more complex spatial relationships. This combination is highly effective for HSI classification tasks.

### 3.2. Attention Mechanism Module

Adding attention modules to deep learning networks can help the network automatically concentrate on significant spectral and spatial features in images, suppress interfering factors, enhance the model’s understanding of spatial context, and effectively capture the edges and structures between ground objects. 

We propose HSIAM, an attention mechanism module comprising spatial, spectral, and channel attention modules. HSIAM typically follows the FFU in the search network. Each processing unit handles comprehensive three-dimensional data, with all three units preserving the interrelationships between spatial and spectral information while focusing on different aspects. This approach optimizes the extraction of relational information between spatial and spectral data. The multi-scale spectral attention module captures spectral information at different resolutions, aiding in capturing details and edges. The multi-scale spatial attention mechanism emphasizes salient features and retains global context. To manage the high computational load of hyperspectral data, which contains numerous bands, we divide the spectral data into smaller blocks for batch processing and introduce channels. The channel attention module processes spatial-spectral data within a single channel. The three submodules of HSIAM effectively fuse different attention outputs through weighted distribution, enhancing feature representation and analysis. Figure 5 illustrates the structure of HSIAM.

In the multi-scale spectral attention module, the input feature map is first subjected to adaptive average pooling to compress the spatial dimension data while preserving spectral information. Next, 3D convolutions with kernel sizes of 1 × 1 × 1, 3 × 3 × 3, and 5 × 5 × 5 are employed to extract local, medium-scale, and large-scale spectral features, respectively. The convolution results at different scales are concatenated in the channel dimension, followed by processing through a 1 × 1 × 1 3D convolution and a ReLU activation function to restore the original number of channels. Finally, normalization is performed by applying a Sigmoid activation function, and an element-wise multiplication with the original input feature map is carried out to obtain enhanced spectral features.

The multi-scale spatial attention module aims to strengthen the spatial information of the input feature map through convolutions of different scales. First, the average and maximum values of the input feature map are calculated along the channel dimension and concatenated. Then, multi-scale spatial feature extraction is performed by combining depth-wise separable convolutions (3 × 3 × 3 kernels) with standard 3D convolutions (3 × 3 × 3 and 5 × 5 × 5 kernels). The results of these multi-scale convolutions are summed and then normalized using a Sigmoid activation function, which is used for element-wise multiplication with the generated spectral feature map, resulting in enhanced spatial features. By combining spectral and spatial attention features, the module achieves complementarity between the two, enhancing the representational capability of features and enabling the model to highlight important features when processing complex inputs more effectively.

The enhanced channel attention module focuses on strengthening the channel information of the input feature map. This module first performs adaptive average pooling and max pooling to compute average and maximum values in the spatial dimension. Following this, a fully connected layer is utilized to reduce and subsequently restore the number of channels while also normalizing the processed features. Ultimately, the normalized feature map is multiplied element-wise with the original input feature map in order to obtain enhanced channel features.

The formula used to calculate the HSIAM is as follows: 

Let x be the input feature map. The output of the multi-scale spectral attention module can be calculated using the following equations:(4)$$c1=concat\left(conv1\left(avg\_pool\left(x\right)\right),conv3\left(avg\_pool\left(x\right)\right),conv5\left(avg\_pool\left(x\right)\right)\right)$$
(5)$${spectral}_{out}=sigmoid\left(ReLU\left(Conv3d\left(c1\right)\right)\right)\cdot x$$
where conv1, conv3, and conv5 represent 3D convolutions with kernel sizes of 1 × 1 × 1, 3 × 3 × 3, and 5 × 5 × 5, respectively, and avg_pool signifies adaptive average pooling. ReLU is the activation function, and Conv3d refers to a 3D convolution with a kernel size of 1 × 1 × 1. Concat denotes concatenation along a specified dimension.

Next, the output calculation formula for the multi-scale spectral-spatial force module is as follows:
(6)$$x_{cat}=concat\left(avg\_out\;,\;max\_out\;\right)$$
(7)$$y1=pointwise\_conv\left(depthwise\_conv\left(x_{cat}\right)\right)$$
(8)$$y2=conv3d\left(y1\right)$$
(9)$$y3=conv3d\left(y2\right)$$
(10)$$y_{sum}=y1+y2+y3$$
(11)$${spatial}_{out}=sigmoid\left(y\_sum\right)\cdot {spectral}_{out}$$
where depthwise_conv signifies depth wise convolution, pointwise_conv denotes pointwise convolution, and max_out is the output of maximum pooling.

The formula for computing the output of the enhanced channel attention module is obtained as follows.

The final output signal of the HSIAM is obtained using:
(12)$${channel}_{out}=\left(fc\left(avg\_pool\left(x\right)\right)+fc\left(max\_pool\left(x\right)\right)\right)\cdot x$$

The final output signal of the HSIAM is obtained using:
(13)$$output=a\cdot {spectral}_{out}+b\cdot {spatial}_{out}+c\cdot {channel}_{out}$$
where a, b, and c represent the weights designated to the spectral attention submodule, spatial attention submodule, and channel attention submodule, respectively. By assigning different weights to the outputs of these three submodules, the HSIAM can better adapt to various tasks.

### 3.3. Loss Function

In HSI classification, selecting a robust and effective loss function is crucial for extracting features from hyperspectral data. Numerous studies [41,42] have demonstrated the effectiveness of integrating different loss functions to address various HSI tasks. The cross-entropy loss (CE loss) [43] function is often employed in HSI classification tasks. While it performs well in many cases, it also exhibits some notable drawbacks. The CE loss focuses on pixel-level classification accuracy, neglecting the spatial relationships between pixels. In HSIs, adjacent pixels often share similar spectral characteristics and categories. Additionally, in hyperspectral data, some classes of samples might be easier to distinguish, while others are relatively difficult [44,45]. For example, identifying land cover types that occupy large contiguous areas is relatively easy, whereas recognizing elongated features such as sidewalks can be challenging. CE loss does not take this aspect into account. It treats all samples equally, which could result in the model optimizing ineffectively and failing to focus on harder-to-distinguish samples. 

To overcome this issue, we propose a Triple Fusion Loss Function (TFLF) composed of CE loss, dice loss [46], and focal loss [47]. Dice loss emphasizes improving the prediction capability for small objects, particularly in scenarios with sample imbalance, and encourages the model to learn minority classes more effectively. It is widely used in classification tasks. Focal Loss, on the other hand, reduces the model’s focus on easy samples and increases the weight on hard-to-classify samples, thereby making the model pay more attention to difficult-to-distinguish samples, such as those at the edges of land cover. By integrating different types of losses, the joint loss function can flexibly adjust the weights for each type of loss based on the features of the data, allowing the model to classify the targets better and facilitating the judgment of land cover boundaries.

The TFLF is defined as follows:(14)$$L_{TFLF}=\alpha L_{CE}+\beta L_{Dice}+\gamma L_{Focal}$$
where *α*, *β*, and *γ* are hyperparameters that balance the contributions of each individual loss component. 

CE loss: This metric is widely used in classification tasks, primarily measuring the difference between two probability distributions. Its formulation is as follows:
(15)$$L_{CE}=-\sum_{i=1}^{N}y_{i}\log\left(p_{i}\right)\;\;$$
where *y_i_* represents the true label and *p_i_* denotes the predicted probability for class *i*.

Dice loss: This loss is a function employed in image segmentation tasks, primarily designed to optimize the F1 score, measuring the similarity between predictions and true labels. It is particularly well-suited for addressing issues associated with class imbalance. The dice loss is defined as:
(16)$$L_{Dice}=1-\frac{2\sum_{i=1}^{N}p_{i}y_{i}+\epsilon }{\sum_{i=1}^{N}p_{i}+\sum_{i=1}^{N}y_{i}+\epsilon }$$
where *ϵ* is a small constant to prevent division by zero, *p_i_* is the predicted probability, and *y_i_* denotes the one-hot encoded true labels.

Focal loss: This loss function mitigates class imbalance by reducing the weight loss assigned to well-classified examples. Its formulation is:
(17)$$L_{Focal}=-\sum_{i=1}^{N}{\left(1-p_{i}\right)}^{\gamma }\log\left(p_{i}\right)\;\;\;$$
where *γ* is a focusing parameter that adjusts the decay rate of the loss weight for easily classified samples.

The TFLF effectively addresses various challenges in HSI classification by combining joint CE loss, dice loss, and focal loss. 

## 4. Experiments

We conducted experiments on a server equipped with an Intel Xeon Gold 6330 CPU 2.00 GHz, 512 GB of memory, and featuring an NVIDIA GeForce RTX 4090 graphics card with 24 GB of video memory (https://www.autodl.com). The deep learning framework utilized was Pytorch 2.3.1. Our validation experiments evaluated the proposed method on three benchmark hyperspectral datasets: MUUFL, Houston2018, and XiongAn. We adopted a rigorous evaluation protocol, randomly selecting a fixed number of labeled pixels per class for training (20 and 30 pixels) and validation (10 pixels), and testing using the remaining pixels. The performance of our method was assessed using three standard metrics, namely, overall accuracy (OA), average accuracy (AA), and Kappa coefficient (Kappa), providing a comprehensive evaluation of classification effectiveness.

### 4.1. Dataset

To assess the practical efficacy of our proposed NAS algorithm, we carefully selected the three representative HSI datasets mentioned above for comparative experiments. These datasets vary by an order of magnitude in sample size, representing small, medium, and large datasets, thereby providing us with a unique perspective to examine the algorithm’s performance from multiple angles. The sample distribution information for these three HSI datasets is listed in Table 2. 

The MUUFL Gulfport dataset [48,49] is an open-source project developed by the GatorSense team at the University of Florida, primarily designed for research in HSI processing and target detection. Collected in November 2010 on the University of Southern Mississippi campus in Long Beach, Mississippi, this dataset incorporates data from various sensors, including hyperspectral and LiDAR data, with a ground sample distance of 1 m. The imagery encompasses a region of 325 by 337 pixels across 72 spectral bands; to mitigate noise interference, the first and last four bands were discarded, resulting in a final compilation of 64 spectral bands, with cropped images measuring 325 by 220 pixels. This dataset features 11 distinct categories, including trees, grasslands, roads, buildings, and various fabric panels, and is regarded as a quintessential small hyperspectral dataset due to its relatively modest size.

The Houston2018 dataset was meticulously gathered at the University of Houston and surrounding urban areas. These hyperspectral data span a spectral range of 380–1050 nm, encompassing 48 distinct bands with a spatial resolution of 1 m. The dataset includes 20 diverse land cover classes featuring roadways, sidewalks, crosswalks, highways, railways, and trains, all exhibiting elongated and narrow shapes that pose significant challenges for classification algorithms. It stands as a quintessential example of a medium-sized hyperspectral dataset.

The XiongAn dataset [50] was collected using a full spectral band multimodal imaging spectrometer integrated into a high-resolution aviation system, which the Shanghai Institute of Technical Physics developed. The spectral range is 400–1000 nm, with 256 bands. The image dimensions are 1580 × 3750 pixels, with a spatial resolution of 0.5 m. The dataset contains 20 land cover types, primarily focused on economic crops. In this dataset, multiple types of economic crops are intermixed within the same area, overlapping each other, and some planting areas are relatively small. This presents a significant challenge for classification models. The XiongAn dataset is a typical large-scale dataset.

### 4.2. Implementation Details

We conducted experiments on two groups to validate the proposed algorithm’s efficacy. In the first group, we randomly selected 20 labeled pixels from each category to construct the training set. An additional 10 labeled pixels were designated for the validation set, with the remaining pixels utilized as the test set. In the second group of experiments, the number of training samples was elevated to 30 for each category, while all other configurations remained consistent with the first group. More details on the distribution of samples are provided in Table 3. 

We constructed three separate search networks with a similar outline structure for the three different datasets. In the external structure, each search network comprises four layers of basic cells, each consisting of three distinct types of cells. Within the internal structure, each operational unit comprises three nodes. Given the storage capacity limitation of 24 GB of VRAM, we cropped patches as search samples at a spatial resolution of 25 × 25 for MUUFL and Houston2018, with a batch size set to 4. For the large-scale XiongAn dataset, 24 GB of VRAM was slightly insufficient, so the patch cropping dimensions were set to 18 × 18, and the batch size was 1. We employed the Adam optimizer for all three datasets to fine-tune the architectural parameters, with the learning rate and weight decay set at 0.001. Concurrently, we utilized the standard SGD optimizer to update the parameters of the search network, specifically configuring the momentum decay at 0.9 and the weight decay at 0.0003. At the same time, the learning rate was systematically reduced from 0.025 to 0.001. For the small MUUFL dataset, the first 16 epochs served as a warm-up phase, during which only the search network parameters were optimized. For the medium-sized Houston2018 dataset, we set 20 epochs for warm-up. For the large-scale XiongAn dataset, the warm-up phase was set to 30 epochs. After the warm-up phase, we updated each iteration’s architecture and search network parameters.

We extracted patches with a spatial resolution measuring 32 × 32 pixels to optimize the network’s training, employing random cropping, flipping, and rotation as data augmentation methods. During the training phase, the batch size for all three datasets was set to 8. At this juncture, we selected the SGD optimizer, equipped with an adjustable learning rate strategy, with an initial learning rate established at 0.1 and decayed according to a multi-learning rate strategy with a power of 0.9. The network’s performance was validated every 100 iterations.

For this task, the weights a, b, and c of the HSIAM were set at 0.4, 0.3, and 0.3, respectively, while the weights α, β, and γ of the TFLF loss function were assigned values of 0.2, 0.5, and 0.3, respectively.

In the inference phase, we employed a sliding window technique to extract small segments, utilizing a stride equivalent to half the window size, which we then input into the training model. The gradient computation was disabled, and the well-trained model was utilized to predict the classification of each pixel within the dataset. We obtained the probability by applying the SoftMax function, selecting the category with the highest probability as the designated pixel class.

### 4.3. Comparative Analysis with Other Techniques

This section compares the proposed TUH-NAS with five other HSI classification methods: SpectralFormer [5], SSFTT [6], GAHT [7], 3D-ANAS [33], and Hyt-NAS [34]. Among these, 3D-ANAS and Hyt-NAS are methods that utilize NAS architectures. All comparative methods use official codes. To ensure fairness, all methodologies employed a consistent training set strategy throughout the experiments: randomly selecting 20 and 30 labeled pixels per class for the training set, 10 labeled pixels for the validation set, and the rest for the test set. Table 4, Table 5, Table 6, Table 7, Table 8 and Table 9 summarize the results of the comparative experiments, and Figure 6, Figure 7 and Figure 8 present the corresponding visual outcomes.

The performance of MUUFL is listed in Table 4 and Table 5. Figure 6 illustrates a comparative analysis of the visual effects yielded by various methods. From the comparative results, we can draw the following conclusions:The proposed TUH-NAS method generally outperforms other comparison methods in classification accuracy when the sample sizes are 20 and 30. For instance, when extracting 20 training samples per class, TUH-NAS achieves an OA of 87.65%, an AA of 88.92%, and a K of 83.90%. This indicates that TUH-NAS has stronger feature extraction and classification capabilities in HSI classification tasks, particularly with smaller sample sizes, where its advantages are more pronounced.Analyzing the specific classification results in both tables, TUH-NAS demonstrates a more balanced performance across different classes, indicating that it is more robust in feature extraction and classification decision-making for all types of samples. This balance is crucial in practical applications as it minimizes overall performance degradation caused by poor classification of certain categories. In contrast, other classification methods may exhibit fluctuations in classification performance for some categories.

Figure 6 displays the visual results when utilizing 30 training samples for each class. A locally zoomed patch is placed on the right side of each result image to illustrate the differences. This locally zoomed patch mainly contains a building (in pink) and its shadow (in purple). From the locally zoomed patch, it can be observed that SpectralFormer misclassifies the entire building as a shadow, while SSFTT identifies parts of the building as a shadow. In GAHT, some building areas are identified as shadows and road. In both 3D-ANAS and Hyt-NAS, some building areas are identified as roads. In the results from TUH-NAS, both the buildings and shadows are correctly identified, with well-defined edges and minimal influence from other classes. Overall, TUH-NAS performs exceptionally well in image detail processing, clearly delineating the edges of trees, grass, roads, and buildings, significantly reducing blurriness and confusion areas. This high clarity aids in the more accurate identification of ground cover. The classification results of TUH-NAS are generally more consistent, with no large-scale misclassifications or outliers.

The comparison results for Houston2018 can be seen in Table 6 and Table 7, as well as Figure 7. The dataset includes more target categories, such as roads, sidewalks, crosswalks, major thoroughfares, highways, railways, and trains. These six elongated shapes are generally more challenging to identify, comprising 30% of the total categories. In this dataset, the classification accuracy of all methods is generally low, with significant discrepancies in accuracy among the various approaches. As illustrated in Table 6 and Table 7, when the training samples per category are limited to 20, our THU-NAS method demonstrates the most superior classification performance across the majority of categories, achieving OA, AA, and Kappa values of 71.51%, 84.52%, and 64.95%, respectively. Conversely, when the training samples per category are increased to 30, both THU-NAS and Hyt-NAS achieve optimal classification for eight categories each, with THU-NAS attaining OA, AA, and Kappa values of 78.47%, 87.79%, and 73.08%, respectively. This indicates that our method exhibits a distinct advantage under conditions of limited training sample conditions. This indicates that our method has an advantage with fewer training samples.

We cropped a small patch from the upper right corner of each result image for magnification, which includes non-residential buildings (dark purple), major thoroughfares (blue), highways (dark red), railways (purple-red), and trains (yellow). Among the methods, SpectralFormer and SSFTT had the poorest identification results, needing more complete recognition of large buildings. GAHT and 3D-ANAS were able to identify buildings to a basic extent but contained some noise. GAHT could hardly distinguish between trains and railways. The 3D-ANAS managed to identify trains and railways but largely failed to recognize major thoroughfares. Hyt-NAS performed well overall but did not identify major thoroughfares. Our proposed TUH-NAS had the best overall classification performance, capable of distinguishing between trains and railways, fully recognizing major thoroughfares and buildings, and exhibiting minimal noise. From the results, it can be seen that, overall, TUH-NAS was better at accurately identifying gaps between aligned roads and accurately depicting road edges. The identification of paved parking lots and cars parked in those lots was also quite accurate. Hyt-NAS also performed well in this aspect, while other classification methods showed poorer results, especially in identifying parking lots in the upper half of the map. TUH-NAS demonstrated the best recognition performance among all methods, with the highest level of discernibility regarding the train on the right side of the image.

The comparative results for XiongAn can be seen in Table 8 and Table 9, as well as Figure 8. The XiongAn dataset includes 20 target categories, primarily consisting of various trees and cultivated crops. The boundaries of major land cover types are quite distinct, and areas of overlapping mixtures occupy a relatively small portion of the entire map. The localized zoomed-in images we extracted mainly include willow (blue), Chinese ash trees (olive green), and lawns (red). The results from the three methods, SpectralFormer, SSFTT, and GAHT, exhibit considerable noise and fail to delineate the boundaries of these three categories clearly. The 3D-ANAS method can identify the boundaries of Chinese ash trees and lawns reasonably well, though there are some noise issues; however, it struggles significantly with the classification of willows. Hyt-NAS can fairly identify the boundaries of all three land cover types, but there are considerable errors in classifying the lawns. Our proposed method, TUH-NAS, accurately identifies all three land cover types with precise boundary delineation and minimal noise. Lawn’s classification performance is the best of all methods. Overall, TUH-NAS and Hyt-NAS show relatively strong classification results, while other classification methods exhibit considerable discrepancies in accuracy. In the left half of the map, Chinese scholar trees (purple), lawns (red), and willows (blue) show significant noise in methods other than TUH-NAS and Hyt-NAS, failing to express the boundaries of each area fully. Hyt-NAS performs well, but there are evident misclassification regions within the Chinese scholar trees and lawn areas. In contrast, TUH-NAS provides a more accurate classification result, fully reflecting the boundary information of each classified land cover. Table 7 and Table 8 show that with 20 training samples per category and 30 classification samples per category, our proposed TUH-NAS obtained superior classification results across the majority of categories. With 20 training samples per category, TUH-NAS’s OA, AA, and Kappa values are 84.36%, 91.11%, and 82.23%, respectively. With 30 training samples per category, the advantages of TUH-NAS become even more pronounced, achieving OA, AA, and K values of 88.95%, 93.14%, and 87.39%, respectively.

### 4.4. Ablation Study

To gain a more profound understanding of the specific impact of each component of the TUH-NAS model on the performance of HSI classification, we embarked on a series of ablation experiments utilizing the Houston 2018 dataset. In these experiments, we selected 30 training samples for each category and meticulously analyzed the effects of individually removing the HSIAM and the TFLF loss function from the TUH-NAS network to assess their contributions to the overall classification efficacy and validate their effectiveness. We established four distinct group configurations. Groups G1, G2, and G3 primarily investigate the roles played by the three constituents of the loss function, while Group G4 focuses on the significance of the attention module, HSIAM.

G1: Removed the HSIAM and the TFLF loss function. The loss function used was CE loss. 

G2: Removed the HSIAM and the TFLF loss function. The loss functions used were CE loss + dice loss. 

G3: Removed the HSIAM. The loss function used was the complete TFLF loss function. 

G4: Included the HSIAM. The loss function used was CE loss instead.

Additionally, we introduced the results from the complete form of TUH-NAS as a reference. The contributions of different modules to the classification task are shown in Table 10.

By comparing the OA, AA, and Kappa coefficients under different configurations, we reached the following conclusions: the simultaneous removal of the attention mechanism module HSIAM and the TFLF loss function and replacement with cross-entropy loss resulted in a significant decline in classification performance. Specifically, the OA, AA, and K values decreased by approximately 4.6%, 1.3%, and 5.1%, respectively. The three-unit mixed search architecture alone achieved a 1% performance improvement over Hyt-NAS. After adding the HSIAM to the mixed search architecture, there was a noticeable enhancement in classification performance, with OA, AA, and K values increasing by about 3.4%, 0.2%, and 3.7%, respectively. This demonstrates that the HSIAM enhances the feature representation capability of input feature maps across spectral, spatial, and channel dimensions, significantly improving the model’s classification accuracy and compared to the sole utilization of the cross-entropy loss function, introducing TFLF as the loss function further augments classification performance. This validates the effectiveness of TFLF in handling HSI classification tasks, as it can better address issues such as boundary recognition in land cover.

### 4.5. Architecture Analysis

The final architectures searched by TUH-NAS are shown in Figure 9. Due to the different spectral and spatial resolutions and the distribution of ground objects across the three datasets, we searched for architectures separately on each dataset. Although these architectures have some unique features in different datasets, they share commonalities in terms of operation types and quantities, which reveal TUH-NAS’s preferences and advantages in dealing with HSI classification tasks. The three architectural diagrams show that the newly introduced dilated convolutions are extensively utilized. In the three datasets, the dilated convolution operation accounts for approximately 25% of the total operations. Additionally, the architecture diagrams for all three datasets reflect that operations related to spectral processing units dominate the final architectures. In the MUUFL and Houston2018 datasets, spectral processing operations account for over 60%, while in the XiongAn dataset, this proportion is close to 50%. These findings indicate that TUH-NAS tends to utilize operations that effectively handle spectral information in HSI classification tasks, while also integrating modern convolution operations and attention mechanisms. This combination demonstrates good performance and adaptability across different datasets.

### 4.6. Comparison of Sensitivity with Different Numbers of Training Sample 

To verify the sensitivity to different training sample sizes, we designed the following experiment using the Houston2018 dataset as the experimental dataset, which contains 20 different categories. We chose the number of samples for each category to be 10, 20, 30, 40, 50, 60, 70, 80, 90, and 100 and tested the model training results in terms of OA, AA, and Kappa for each sample size. We limited the maximum number of training iterations to 100,000 to save time. Other settings remained consistent with the previous experimental scheme. The final experimental results are shown in Table 11. We represented the data in Table 11 using a line chart, as shown in Figure 10.

From Table 11 and Figure 10, we can analyze the sensitivity of the experimental results to different sample sizes:Overall Accuracy (OA):

The OA increases consistently as the number of training samples per class increases from 10 to 100. This suggests that a larger training sample size improves the model’s overall performance on the classification task, likely due to more data providing better representation and variability.

There is a noticeable jump from 57.93% OA at 10 samples to 73.00% at 20 samples, indicating that even a modest increase in sample size can lead to significant gains in performance.

2.Average Accuracy (AA):

The AA also shows a steady increase with the number of samples. The AA starts at 79.49% with 10 samples and approaches 92.99% with 100 samples. This indicates a strong sensitivity of average class accuracy to the increase in training samples, which might reflect better differentiation between classes as more examples are available to learn from.

3.Kappa Index (K × 100):

The Kappa index also increases as the sample size grows, from 50.84 at 10 samples to 82.32 at 100 samples. Kappa provides a measure of agreement between predicted classifications and actual classes while accounting for chance agreements. The substantial increase in Kappa, along with the increase in sample size, indicates improved model reliability and consistency in classifications.

Overall, the consistent improvements in OA, AA, and Kappa with increased sample sizes reflect the sensitivity of the classification model’s performance to the quantity of training data available. It suggests that larger sample sizes lead to better performance metrics in HSI classification tasks, likely due to enhanced model training and reduced overfitting. This analysis underscores the importance of adequate sample sizes in machine learning tasks to achieve robust model performance.

### 4.7. About Training Time and Number of Model Parameters

We have recorded the average execution time of six comparison methods on the same device across three datasets, as shown in Table 12. We also obtained the number of trainable parameters in the actual models for six different methods when processing the three datasets, as shown in Table 13. For NAS networks, the execution time is the sum of the time spent searching the network architecture and the time spent on model optimization training.

From the average execution times shown in Table 12, there are significant differences in the running times of different deep learning models on the hyperspectral classification task. The three Transformer-based methods, SpectralFormer, SSFTT, and GAHT, demonstrate significantly lower average execution times across the three datasets compared to the latter three NAS-based methods. However, running time is not the only metric for assessing model performance. Despite the high efficiency of SSFTT and GAHT, they do not achieve the same classification accuracy as the NAS-based models. Among the three NAS-based methods, although TUH-NAS has the highest model complexity, thanks to our optimization of the degree code and the use of mixed precision during architecture search, its average execution time is not the highest.

From Table 13, we can observe clear differences in the model parameters of the three Transformer-based methods across different datasets, while the model parameters of the three NAS-based methods remain relatively stable. This is due to the varying architectural designs and module complexities. The parameter count of Transformer-based models is typically closely related to the number of layers, the number of neurons per layer, and the implementation of the self-attention mechanism. Different models may adopt different layer encoding mechanisms, numbers of attention heads, and other hyperparameters, leading to significant variations in parameter counts for the same task. In contrast, NAS-based methods identify optimal model structures through an automated NAS process, which tends to be more standardized and uniform. As a result, the generated models exhibit a more consistent number of parameters, and the searched and selected models usually strike a good balance between complexity and performance. This characteristic of architectural automation allows NAS models to maintain relatively consistent parameter scales when facing different datasets.

Overall, TUH-NAS presents itself as a viable model choice for HSI classification tasks, with a moderate and relatively stable number of parameters alongside good execution time performance. It is suitable for practical applications where resource consumption and classification performance need to be considered comprehensively.

## 5. Conclusions

In this paper, a new triple-unit NAS network framework called TUH-NAS is proposed to address the issues of existing NAS-based classification methods that struggle to effectively combine spectral information and spatial information, as well as the potential loss of global contextual information. TUH-NAS includes three working units: SPEU, SPAU, and FFU, where SPEU focuses on deep spectral information mining, SPAU emphasizes spatial feature extraction, and FFU is responsible for merging the information from the first two units to strengthen the connection between spectral and spatial dimensions. Under the TUH-NAS framework, we also introduce a new HSIAM to enhance sensitivity to critical features such as edge regions. Moreover, we employ a comprehensive loss function TFLF, which consists of three joint loss functions to increase the model’s focus on challenging samples, such as those at object edges, thereby further improving overall classification performance. Despite its promising performance, TUH-NAS faces challenges in terms of computational efficiency and scalability to larger datasets. The complexity of the triple-unit architecture and the attention mechanism module HSIAM introduce substantial computational overhead, making it less suitable for real-time applications or processing of very large-scale HSIs. In future research, we will optimize computational efficiency and improve the model’s generalization ability to meet the demands of more complex scenarios.

## Figures and Tables

**Figure 1 sensors-24-07834-f001:**
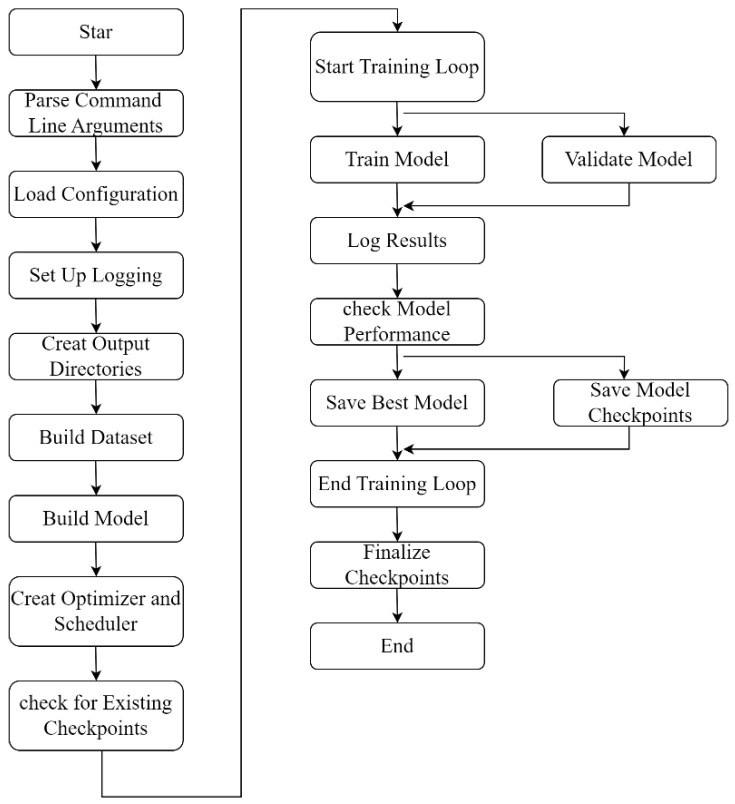
The overall algorithm flowchart of the proposed method.

**Figure 2 sensors-24-07834-f002:**
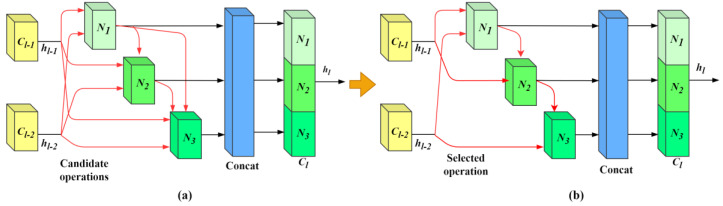
Internal search architecture and search results. (**a**) illustrates the structure of the basic search cell. The red arrows represent the set of all candidate operations. (**b**) represents the structure search results, where each node retains only the two most valuable input paths. The red arrows represent the selected basic operations.

**Figure 3 sensors-24-07834-f003:**
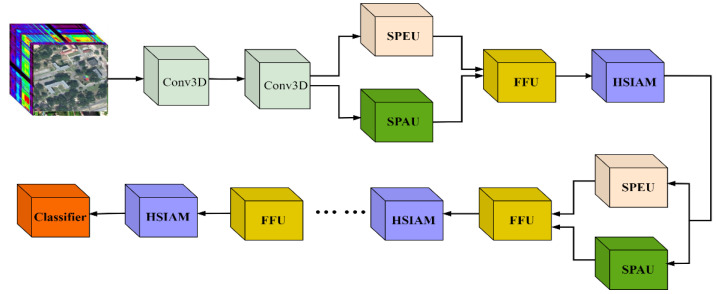
External search network framework.

**Figure 4 sensors-24-07834-f004:**
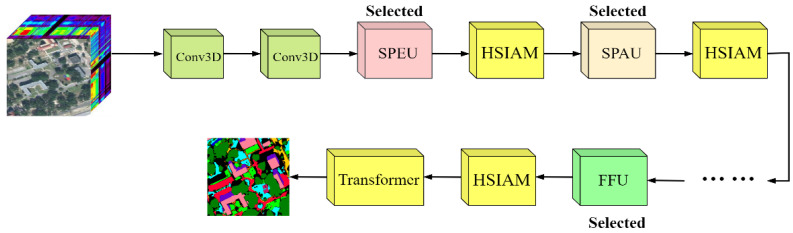
The final optimized training network framework.

**Figure 5 sensors-24-07834-f005:**
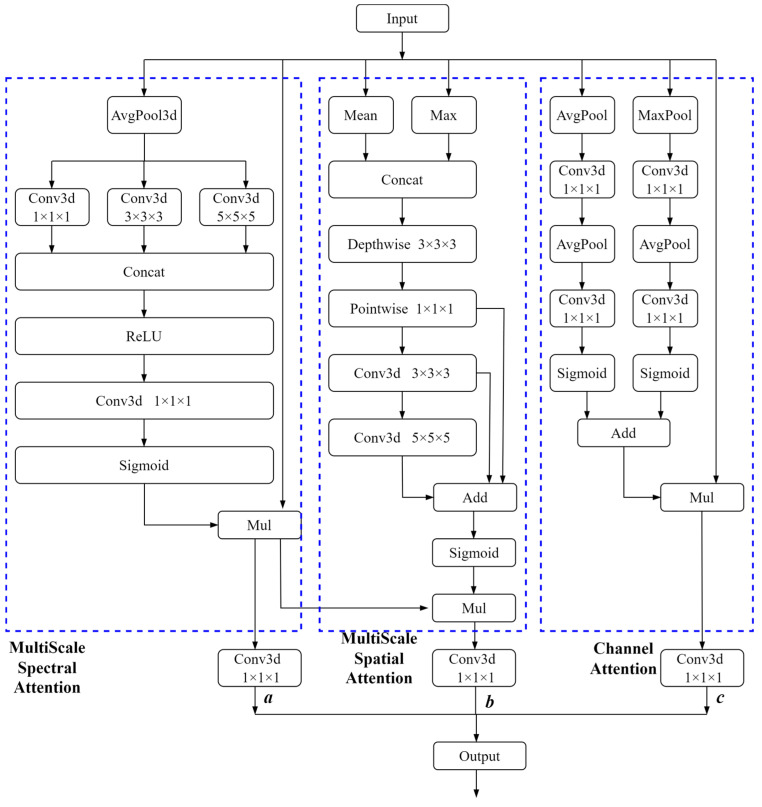
Diagram of the HSIAM structure.

**Figure 6 sensors-24-07834-f006:**
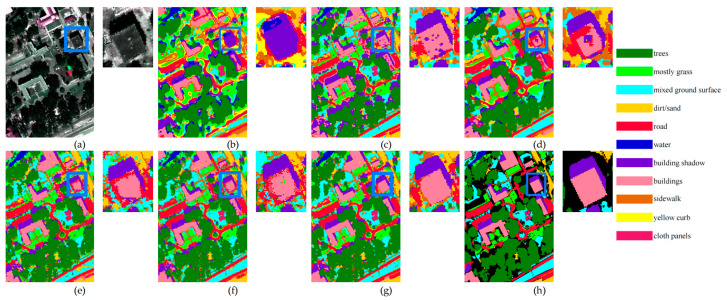
Comparison of experimental results on MUUFL with 30 training samples per class. (**a**) False color composite image; (**b**) SpectralFormer; (**c**) SSFTT; (**d**) GAHT; (**e**) 3-D-ANAS; (**f**) Hyt-NAS; (**g**) TUH-NAS; (**h**) Ground-truth map.

**Figure 7 sensors-24-07834-f007:**
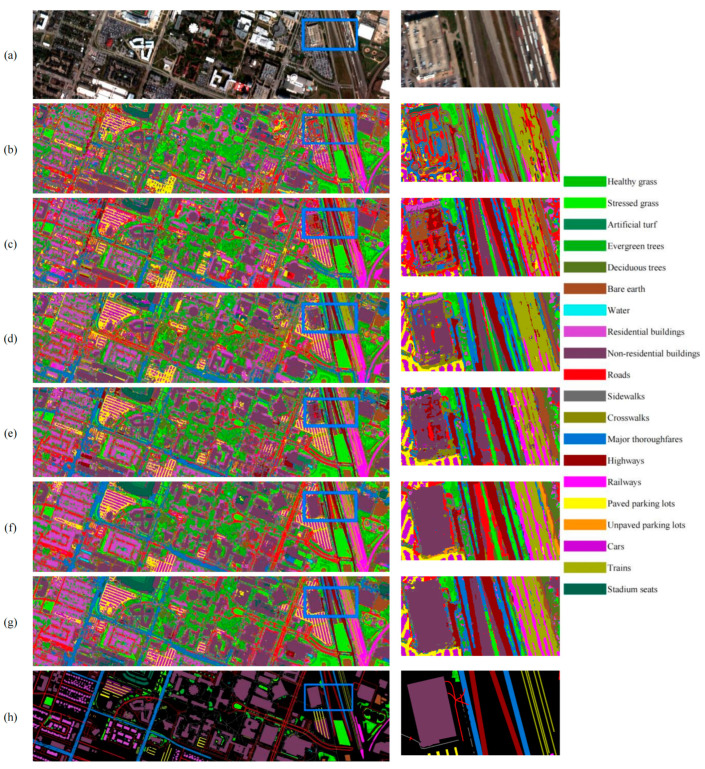
Comparison of experimental results on Houston2018 with 30 training samples per class. (**a**) False color composite image; (**b**) SpectralFormer; (**c**) SSFTT; (**d**) GAHT; (**e**) 3-D-ANAS; (**f**) Hyt-NAS; (**g**) TUH-NAS; (**h**) Ground-truth map.

**Figure 8 sensors-24-07834-f008:**
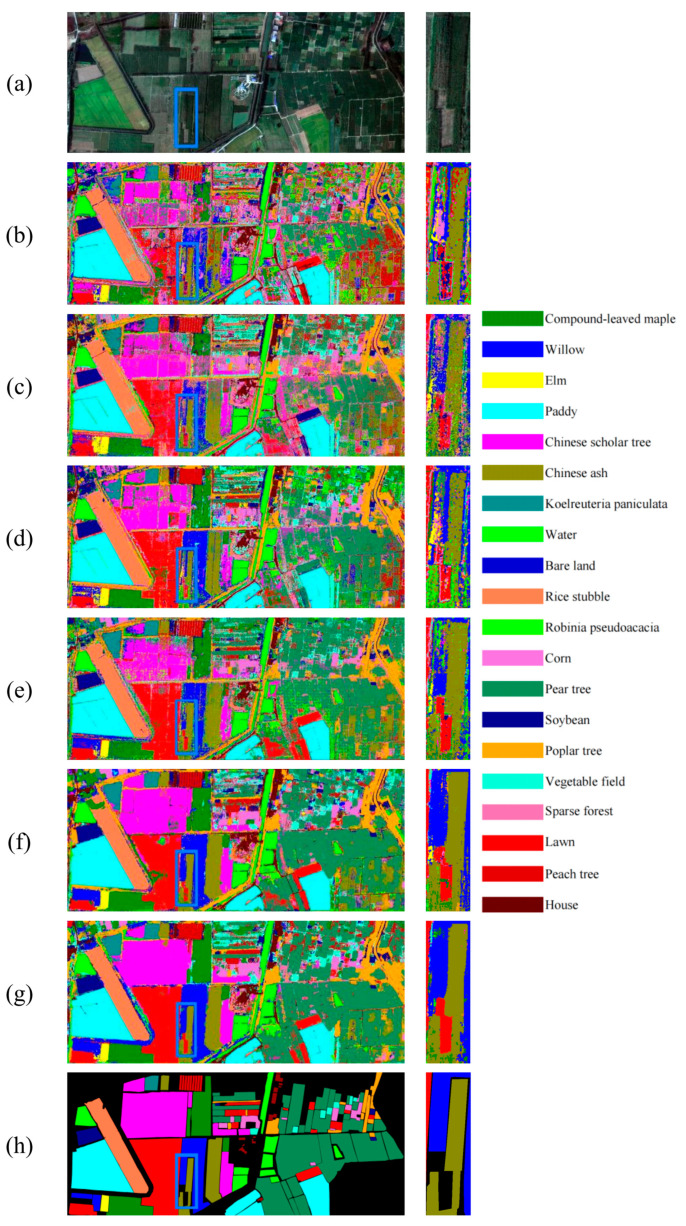
Comparison of experimental results on XiongAn with 30 training samples per class. (**a**) False color composite image; (**b**) SpectralFormer; (**c**) SSFTT; (**d**) GAHT; (**e**) 3-D-ANAS; (**f**) Hyt-NAS; (**g**) TUH-NAS; (**h**) Ground-truth map.

**Figure 9 sensors-24-07834-f009:**
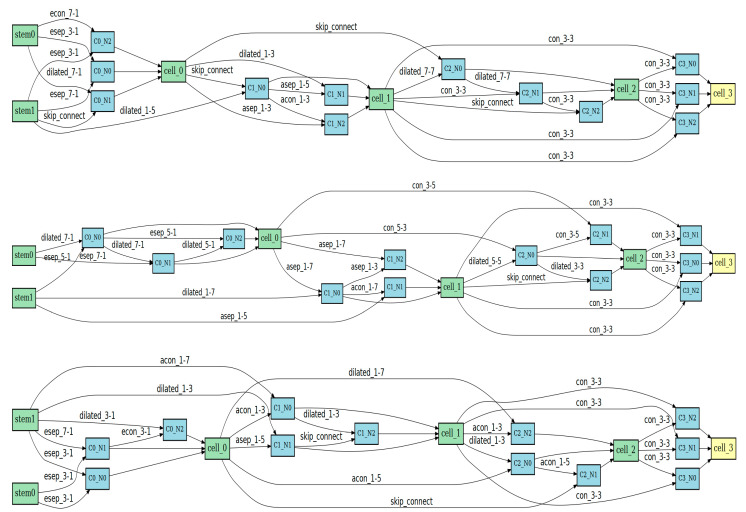
Final architectures found for three datasets. From top to bottom: MUUFL, Houston2018, and XiongAn. The green nodes are the main working units of the search network, the yellow nodes are the output nodes, and the blue nodes are the sub-nodes within the working units.

**Figure 10 sensors-24-07834-f010:**
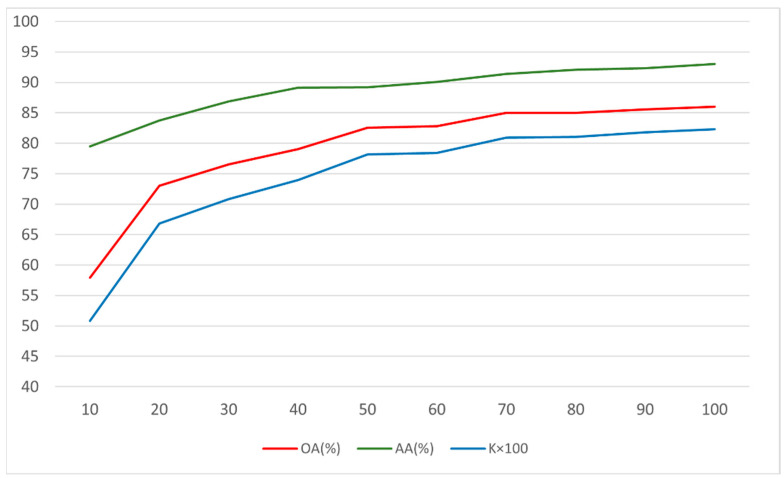
Line chart corresponding to the data in Table 11.

**Table 1 sensors-24-07834-t001:** The candidate operations included in each unit.

Spectral Cell		Spatial Cell		Fusion Cell	
No.	Candidate Operation	No.	Candidate Operation	No.	Candidate Operation
1	econ_3-1	1	acon_1-3	1	con_3-3
2	econ_5-1	2	acon_1-5	2	con_3-5
3	econ_7-1	3	acon_1-7	3	con_5-3
4	esep_3-1	4	asep_1-3	4	con_5-5
5	esep_5-1	5	asep_1-5	5	con_5-7
6	esep_7-1	6	asep_1-7	6	con_7-7
7	dilated_3-1	7	dilated_1-3	7	dilated_3-3
8	dilated_5-1	8	dilated_1-5	8	dilated_5-5
9	dilated_7-1	9	dilated_1-7	9	dilated_7-7
10	skip_connect	10	skip_connect	10	skip_connect
11	none	11	none	11	none

**Table 2 sensors-24-07834-t002:** Sample distribution information of the datasets.

	MUUFL		Houston2018		XiongAn	
Class	Land Cover	Samples Size	Land Cover	Sample Size	Land Cover	Sample Size
1	Trees	23,246	Healthy grass	9799	Compound-leaved maple	225,647
2	Mostly grass	4270	Stressed grass	32,502	Willow	180,766
3	Mixed ground surface	6882	Artificial turf	684	Elm	15,353
4	Dirt/sand	1826	Evergreen trees	13,595	Paddy	452,144
5	Road	6687	Deciduous trees	5021	Chinese scholar tree	475,591
6	Water	466	Bare earth	4516	Chinese ash	169,342
7	Building shadow	2233	Water	266	Koelreuteria paniculata	23,304
8	Buildings	6240	Residential buildings	39,772	Water	165,647
9	Sidewalk	1385	Non-residential buildings	223,752	Bare land	38,409
10	Yellow curb	183	Roads	45,866	Rice stubble	193,830
11	Cloth panels	269	Sidewalks	34,029	Robinia pseudoacacia	5612
12			Crosswalks	1518	Corn	59,165
13			Major thoroughfares	46,348	Pear tree	1,026,513
14			Highways	9865	Soybean	7151
15			Railways	6937	Poplar tree	91,072
16			Paved parking lots	11,500	Vegetable field	29,148
17			Unpaved parking lots	146	Sparse forest	1496
18			Cars	6547	Lawn	421,790
19			Trains	5369	Peachtree	65,514
20			Stadium seats	6824	House	29,616
	Total	53,687	Total	504,856	Total	3,677,110

**Table 3 sensors-24-07834-t003:** Distribution information for the training, validation, and test sets.

Setting	Dataset	Training	Validation	Test	Training%
20 pixels/class	MUUFL	220	110	53,357	0.409%
	Houston2018	400	200	504,256	0.079%
	XiongAn	400	200	3,676,510	0.011%
30 pixels/class	MUUFL	330	110	53,247	0.615%
	Houston2018	600	200	504,056	0.112%
	XiongAn	600	200	3,676,310	0.016%

**Table 4 sensors-24-07834-t004:** Comparison experiment results of 20 training samples per class in MUUFL.

Class	SpectralFormer	SSFTT	GAHT	3D-ANAS	Hyt-NAS	TUH-NAS
1	70.39	81.98	83.34	88.57	87.54	**91.55**
2	76.93	79.41	74.53	85.87	87.33	**87.62**
3	42.14	56.36	69.59	74.05	75.19	**79.26**
4	80.34	**95.71**	86.53	88.03	90.70	89.59
5	62.76	82.48	81.03	87.82	90.30	**91.00**
6	97.42	85.32	**100.00**	97.94	99.77	99.31
7	35.78	**91.83**	90.20	85.93	89.61	87.34
8	57.36	73.77	75.57	**79.00**	75.78	78.97
9	38.63	**81.25**	72.62	74.98	71.59	77.86
10	67.21	77.12	76.47	**100.00**	98.04	96.08
11	97.77	99.58	98.33	99.58	**100.00**	99.58
OA (%)	63.27	78.54	79.98	84.97	84.88	**87.65**
AA (%)	66.07	82.26	82.56	87.43	87.80	**88.92**
K × 100	54.90	72.75	74.59	80.59	80.45	**83.90**

**Table 5 sensors-24-07834-t005:** Comparison experiment results of 30 training samples per class in MUUFL.

Class	SpectralFormer	SSFTT	GAHT	3D-ANAS	Hyt-NAS	TUH-NAS
1	77.60	81.71	86.04	86.87	87.76	**91.39**
2	78.17	**85.74**	76.10	79.69	76.83	81.70
3	33.16	72.65	74.45	75.02	**81.94**	81.54
4	61.66	85.72	87.96	**94.46**	90.71	90.76
5	63.87	78.47	82.55	89.21	90.49	**92.61**
6	96.57	93.66	**100.00**	98.59	97.89	99.53
7	63.64	95.26	81.40	91.02	92.52	**96.22**
8	65.54	80.77	81.65	88.13	85.52	**92.19**
9	51.55	77.03	85.58	76.13	**88.62**	82.38
10	71.58	89.51	76.22	97.20	97.90	**99.30**
11	97.40	94.32	95.63	99.13	99.13	**99.13**
OA (%)	67.29	81.10	82.80	85.55	86.70	**89.67**
AA (%)	69.16	84.99	84.33	88.68	89.94	**91.52**
K × 100	59.27	75.98	77.95	81.35	82.80	**86.54**

**Table 6 sensors-24-07834-t006:** Comparison experiment results of 20 training samples per class in Houston2018.

Class	SpectralFormer	SSFTT	GAHT	3-D-ANAS	Hyt-NAS	TUH-NAS
1	90.92	78.67	66.85	**94.91**	86.81	93.24
2	70.48	**88.25**	87.50	80.90	88.04	85.86
3	94.01	99.39	99.08	99.53	**100.00**	**100.00**
4	88.83	91.94	92.68	96.71	89.51	**96.89**
5	53.91	86.92	85.75	80.49	85.09	**87.26**
6	82.77	86.91	94.27	94.71	98.93	**99.58**
7	83.08	96.19	99.58	**100.00**	**100.00**	**100.00**
8	63.39	66.00	71.11	72.08	81.99	**86.77**
9	34.24	51.04	51.37	71.28	66.58	**74.33**
10	14.18	15.46	36.33	23.34	39.29	**44.36**
11	19.12	49.12	35.72	**45.89**	43.76	44.15
12	37.42	49.13	31.99	52.84	42.34	**61.29**
13	23.76	29.12	32.83	**44.59**	44.10	40.49
14	51.30	79.26	84.82	81.21	91.50	**93.09**
15	80.67	98.19	91.69	97.40	97.25	**99.00**
16	51.30	72.12	85.44	86.66	87.99	**95.40**
17	**100.00**	98.28	100.00	96.23	**100.00**	**100.00**
18	66.70	87.52	77.78	74.26	**92.77**	88.71
19	40.96	77.58	76.81	95.06	95.84	**99.96**
20	90.61	84.66	91.33	92.51	99.16	**100.00**
OA (%)	41.01	54.52	56.43	66.36	67.05	**71.51**
AA (%)	61.88	74.29	74.65	79.03	81.55	**84.52**
K × 100	33.90	47.24	49.17	58.69	59.96	**64.95**

**Table 7 sensors-24-07834-t007:** Comparison experiment results of 30 training samples per class in Houston 2018.

Class	SpectralFormer	SSFTT	GAHT	3-D-ANAS	Hyt-NAS	TUH-NAS
1	83.53	78.34	93.62	90.72	82.32	90.87
2	84.40	89.47	75.10	82.61	**86.90**	84.67
3	98.91	99.84	100.00	99.84	**100.00**	**100.00**
4	93.08	88.48	86.25	96.21	**98.41**	97.12
5	70.39	82.67	88.18	93.41	92.63	**94.22**
6	87.78	97.05	93.95	99.80	92.18	**100.00**
7	98.67	97.79	**100.00**	**100.00**	99.56	99.56
8	54.32	74.85	76.28	85.73	91.66	**92.36**
9	38.08	51.05	56.16	68.34	68.93	**81.71**
10	23.12	38.98	26.37	46.43	50.25	**52.43**
11	24.53	36.79	40.58	42.90	**54.41**	52.97
12	34.03	53.25	61.71	52.77	**67.19**	63.94
13	17.64	42.45	47.59	55.49	**63.31**	60.29
14	67.54	90.13	83.48	91.70	**97.27**	94.75
15	91.66	96.88	92.84	98.13	**100.00**	99.78
16	65.02	87.29	88.64	94.99	**97.29**	95.51
17	**100.00**	**100.00**	**100.00**	**100.00**	**100.00**	**100.00**
18	68.71	84.00	90.61	81.19	94.81	**96.17**
19	75.66	82.10	96.83	95.89	98.05	**99.53**
20	90.39	90.18	83.21	94.60	99.96	**100.00**
OA (%)	44.83	58.40	59.72	69.75	72.91	**78.47**
AA (%)	68.37	78.08	79.07	83.54	86.76	**87.79**
K × 100	37.49	51.30	52.71	63.29	67.08	**73.08**

**Table 8 sensors-24-07834-t008:** Comparison experiment results of 20 training samples per class in XiongAn.

Class	SpectralFormer	SSFTT	GAHT	3D-ANAS	Hyt-NAS	TUH-NAS
1	34.69	44.01	74.15	54.83	74.88	**83.72**
2	45.88	46.70	70.29	81.14	90.56	**91.82**
3	91.25	95.76	99.15	93.33	**99.80**	98.89
4	88.25	83.82	93.78	92.36	95.35	**95.40**
5	28.53	30.94	56.07	49.00	81.64	**85.49**
6	52.09	75.06	82.47	75.79	93.53	**95.68**
7	89.68	95.63	94.25	92.09	**99.45**	97.98
8	85.00	88.19	89.05	85.49	95.44	**95.51**
9	90.47	89.08	91.67	97.86	99.67	**99.83**
10	90.81	92.30	93.85	85.12	**98.36**	96.53
11	68.41	57.00	91.88	80.29	99.75	**99.95**
12	35.43	58.22	71.66	76.21	**94.37**	93.34
13	31.10	44.32	40.14	53.97	68.30	**71.46**
14	77.88	63.63	85.35	93.01	**99.27**	99.13
15	53.29	34.06	65.21	67.21	73.13	**74.64**
16	36.28	51.47	58.13	58.14	**90.06**	83.04
17	52.87	63.37	78.99	91.00	**99.25**	96.04
18	46.77	52.44	70.27	58.12	79.25	**82.02**
19	47.45	67.78	67.14	68.54	**85.65**	85.43
20	77.93	85.74	86.39	81.93	93.93	**96.40**
OA (%)	49.81	56.25	66.24	66.18	82.08	**84.36**
AA (%)	61.20	65.98	77.99	76.77	90.58	**91.11**
K × 100	45.18	51.62	62.65	62.23	79.73	**82.23**

**Table 9 sensors-24-07834-t009:** Comparison experiment results of 30 training samples per class in XiongAn.

Class	SpectralFormer	SSFTT	GAHT	3D-ANAS	Hyt-NAS	TUH-NAS
1	52.25	51.98	79.21	71.37	86.00	**90.21**
2	44.64	54.01	72.16	74.68	93.77	**95.14**
3	93.78	93.92	98.14	95.74	99.86	**99.82**
4	90.16	93.51	95.39	94.71	97.31	**98.01**
5	42.59	60.79	66.10	61.33	90.36	**92.24**
6	59.32	54.86	69.63	82.39	**95.89**	93.66
7	89.96	97.32	97.19	99.32	99.89	**99.99**
8	86.25	91.14	95.41	85.70	**96.78**	89.95
9	92.36	97.26	93.73	93.94	99.92	**99.96**
10	93.81	95.05	96.31	96.62	92.32	**98.25**
11	69.67	71.84	96.40	95.60	99.37	**99.98**
12	59.77	77.16	80.91	81.87	**91.15**	83.92
13	35.73	38.78	54.84	62.84	71.98	78.59
14	79.12	82.87	89.96	90.70	**99.62**	97.60
15	40.55	71.58	79.74	72.28	**89.68**	85.32
16	46.76	63.70	70.78	55.56	80.71	**85.50**
17	85.65	84.82	91.90	96.43	99.73	**100.00**
18	42.38	56.46	67.47	76.82	79.37	**89.76**
19	51.53	59.18	78.85	77.56	86.28	**89.32**
20	83.20	81.41	82.12	84.44	**96.26**	95.56
OA (%)	54.54	61.71	72.57	74.71	85.50	**88.95**
AA (%)	66.97	73.88	82.31	82.50	92.31	**93.14**
K × 100	50.06	57.77	69.38	71.50	83.57	**87.39**

**Table 10 sensors-24-07834-t010:** Comparison experiment results of 30 training samples per class in XiongAn.

State	C1	C2	C3	C4	TUH-NAS
OA (%)	73.83	74.02	74.49	77.23	78.47
AA (%)	86.52	86.64	86.47	86.68	87.79
K × 100	67.96	67.53	68.67	71.69	73.08

**Table 11 sensors-24-07834-t011:** OA, AA, and Kappa for different numbers of training samples under the Houston2018 dataset.

	10	20	30	40	50	60	70	80	90	100
OA (%)	57.93	73.00	76.51	79.02	82.56	82.82	84.97	84.97	85.58	86.01
AA (%)	79.49	83.72	86.86	89.14	89.19	90.07	91.38	92.10	92.35	92.99
K × 100	50.84	66.83	70.82	73.96	78.18	78.41	80.90	81.03	81.78	82.32

**Table 12 sensors-24-07834-t012:** Statistical table of average execution time for 6 methods.

Model	SpectralFormer	SSFTT	GAHT	3D-ANAS	Hyt-NAS	TUH-NAS
MUUFL	1 m 11 s	34 s	58 s	3 h 21 m 17 s	8 h 47 m 42 s	6 h 51 m 23 s
Houston2018	5 m 53 s	2 m 7 s	3 m 31 s	5 h 9 m 11 s	5 h 36 m 28 s	7 h 07 m 30 s
XiongAn	36 m 20 s	14 m 56 s	20 m 54 s	9 h 48 m 39 s	29 h 16 m 02 s	24 h 57 m 34 s

**Table 13 sensors-24-07834-t013:** The number of parameters for each model.

Model	3D-ANAS	Hyt-NAS	TUH-NAS
MUUFL	243.96 K	230.38 K	398.28 K
Houston2018	243.36 K	234.77 K	368.31 K
XiongAn	237.40 K	232.56 K	312.48 K

## Data Availability

The MUUFL dataset can be obtained at the url: https://github.com/GatorSense/MUUFLGulfport.git (accessed on 28 June 2023). The Hoouston2018 dataset can be obtained at the URL: https://machinelearning.ee.uh.edu/2018-ieee-grss-data-fusion-challenge-fusion-of-multispectral-lidar-and-hyperspectral-data/ (accessed on 11 May 2023). The XiongAn dataset can be freely available at the URL: http://www.hrs-cas.com/a/share/shujuchanpin/2019/0501/1049.html (accessed on 18 March 2023).

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
