# Peer review of "TUH-NAS: A Triple-Unit NAS Network for Hyperspectral Image Classification"

_sensors, 2024, doi:10.3390/s24237834_

Round 1
Reviewer 1 Report
Comments and Suggestions for Authors
Technology:
1. A Triple-Unit Hyperspectral NAS network is proposed to address the issue of existing NAS-based hyperspectral image classification methods not paying special attention to the complex relationship between spectral data and spatial data.
2. HSIAM attention mechanism module.
3. Triple Fusion Loss Function.
Suggestions:
1. Proper nouns and abbreviations require further careful examination. For example, in line 27, the abbreviation for hyperspectral image should be HSI.
2. Machine learning-based hyperspectral classification methods combining spectral and spatial have still been developed in recent years:
doi: 10.1109/TGRS.2024.3380087
doi: 10.1109/JIOT.2024.3412925
Such methods should be analysed comprehensively to ensure the completeness of the literature research.
3. In the fourth contribution, quantitative conclusions drawn from comparative experiments should be emphasised.
4. The presentation of the neural architecture search and attention mechanisms in related work should be simplified, focusing on their algorithmic principles and their role in hyperspectral image classification.
5. The preface to the experiment (line 504) should indicate which validation experiments were completed and the outcome assessment metrics used.
6. Experiments on the sensitivity of the proposed method to different numbers of training samples need to be supplemented (Please compare OA and Kappa for all comparison methods).
7. The average execution time of all methods on the same device needs to be compared.
8. The limitations of TUH-NAS in the conclusions should be further briefly described.
Comments on the Quality of English LanguageThe quality of English should be improved.
Reviewer 2 Report
Comments and Suggestions for Authors
1. Rewrite 'Over the last few years, neural architecture search (NAS) technology has 9 achieved good results in hyperspectral'. This is not clear enough.
2. From lines 92 to 128 you have not cited any reference. please cite appropriate references.
3. write your list of contributions concisely.
4. Keep your related work concise and specific. In the section '2.1 Neural Architecture Search,' rewrite and explain related work so the audience can understand the relevant methods. Also, cite appropriate references.
5. Remove duplicate information from the introduction and related method. also, Remove all redundant information from the manuscript, making it concise and informative.
6. Clearly mention the problem of the existing method and How your model solves the problem.
7. Rewrite your conclusion by highlighting your findings to solve your objectives.
8. Remove your proposed attention model from section 2.2 and move to the methodology section.
9. In Figure 1, use consistent formatting.
10. use appropriate reference for the statement between lines 462 to 466. Also, cite references for the three losses of Cross-Entropy Loss, Dice Loss, and Focal Loss.
Reviewer 3 Report
Comments and Suggestions for Authors
The paper presents a novel neural architecture search (NAS) network, Triple-unit Hyperspectral NAS (TUH-NAS), for hyperspectral image classification. TUH-NAS emphasizes the integration of spatial and spectral features, which is critical for improving classification accuracy. A new hyperspectral image attention mechanism module (HSIAM) is introduced to focus on important regions and enhance sensitivity to priority areas. Additionally, a composite loss function is used to address challenging samples. Experimental results on three public hyperspectral datasets show that TUH-NAS outperforms existing NAS methods, particularly in recognizing object edges, even with a limited number of samples. The overall structure of the paper is reasonable, the formulas are correct, and the experimental results are substantial and credible. The reviewers have the following minor issues:
(1) There are several instances of incorrect acronyms in the text, such as HSI written as HIS. Please check the entire document and make corrections.
(2) It is suggested to add an algorithm flowchart.
(3) It is recommended to cite and briefly discuss related work: doi: 10.1109/TGRS.2024.3392264,and 10.1109/TGRS.2022.3208897.
(4) A brief analysis of the proposed method’s running efficiency and the number of parameters is suggested.
Round 2
Reviewer 1 Report
Comments and Suggestions for Authors
Most of concerns have been addressed nu authors, and it can be accepted after carefully check.
Reviewer 2 Report
Comments and Suggestions for Authors
If possible check plagiarism and inappropriate citations
